# Immune Checkpoint Blockade via PD-L1 Potentiates More CD28-Based than 4-1BB-Based Anti-Carbonic Anhydrase IX Chimeric Antigen Receptor T Cells

**DOI:** 10.3390/ijms23105448

**Published:** 2022-05-13

**Authors:** Najla Santos Pacheco de Campos, Adriano de Oliveira Beserra, Pedro Henrique Barbosa Pereira, Alexandre Silva Chaves, Fernando Luiz Affonso Fonseca, Tiago da Silva Medina, Tiago Goss dos Santos, Yufei Wang, Wayne Anthony Marasco, Eloah Rabello Suarez

**Affiliations:** 1Center for Natural and Human Sciences, Federal University of ABC, Santo Andre 09210-580, SP, Brazil; naj.pacheco@gmail.com; 2A.C. Camargo Cancer Center, Centro Internacional de Pesquisa, Sao Paulo 01508-010, SP, Brazil; adriano.beserra@accamargo.org.br (A.d.O.B.); pedro.pereira@accamargo.org.br (P.H.B.P.); alexandre.chaves@accamargo.org.br (A.S.C.); tiago.medina@accamargo.org.br (T.d.S.M.); tsantos@accamargo.org.br (T.G.d.S.); 3Laboratório de Análises Clínicas, Centro Universitário Faculdade de Medicina do ABC, Santo Andre 09060-870, SP, Brazil; profferfonseca@gmail.com; 4Departamento de Ciências Farmacêuticas, Universidade Federal de Aso Paulo, Diadema 09920-000, SP, Brazil; 5Department of Cancer Immunology and Virology, Dana-Farber Cancer Institute, Boston, MA 02215, USA; yufei_wang@dfci.harvard.edu; 6Department of Medicine, Harvard Medical School, Boston, MA 02215, USA

**Keywords:** adoptive T cell therapy, hypoxic tumors, immune checkpoint blockade, CAR T, solid tumors, T cell exhaustion, co-stimulatory domains, CD28, 4-1BB, CD137

## Abstract

The complete regression of clear cell renal cell carcinoma (ccRCC) obtained pre-clinically with anti-carbonic anhydrase IX (CAIX) G36 chimeric antigen receptor (CAR) T cells in doses equivalent to ≅10^8^ CAR T cells/kg renewed the potential of this target to treat ccRCC and other tumors in hypoxia. The immune checkpoint blockade (ICB) brought durable clinical responses in advanced ccRCC and other tumors. Here, we tested CD8α/4-1BB compared to CD28-based anti-CAIX CAR peripheral blood mononuclear cells (PBMCs) releasing anti-programmed cell death ligand-1 (PD-L1) IgG4 for human ccRCC treatment in vitro and in an orthotopic NSG mice model in vivo. Using a ≅10^7^ CAR PBMCs cells/kg dose, anti-CAIX CD28 CAR T cells releasing anti-PD-L1 IgG highly decrease both tumor volume and weight in vivo, avoiding the occurrence of metastasis. This antitumoral superiority of CD28-based CAR PBMCs cells compared to 4-1BB occurred under ICB via PD-L1. Furthermore, the T cell exhaustion status in peripheral CD4 T cells, additionally to CD8, was critical for CAR T cells efficiency. The lack of hepatotoxicity and nephrotoxicity upon the administration of a 10^7^ CAR PMBCs cells/kg dose is the basis for carrying out clinical trials using anti-CAIX CD28 CAR PBMCs cells releasing anti-PD-L1 antibodies or anti-CAIX 4-1BB CAR T cells, offering exciting new prospects for the treatment of refractory ccRCC and hypoxic tumors.

## 1. Introduction

T cell exhaustion arises from the constant antigenic stimulation of T cell receptors and the frequent overexpression of immune checkpoint-inducing molecules, e.g., programmed cell death ligand-1 (PD-L1), within the tumor microenvironment (TME), leading to the progressive loss of T cell effector functions [1]. The immune checkpoint blockade (ICB) targeting the programmed cell death receptor-1 (PD-1) and PD-L1 axis alone or in association with the cytotoxic T-lymphocyte-associated protein 4 (CTLA4) blockade brought durable clinical responses for clear cell renal cell carcinoma (ccRCC) in adjuvant settings and metastatic scenarios, becoming an important pillar treatment [2,3,4].

Besides ICB, chimeric antigen receptor (CAR) T cells are an effective form of adoptive cell therapy designed against tumor antigens that have demonstrated remarkable effects for the treatment of hematological tumors [5]. For CD19+ tumors, CD28 or 4-1BB-based second-generation CARs are by far the most applied. CD28 incorporation into the anti-CD19 CAR structure promotes effector memory maturation, glycolysis, rapid tumor eradication, and T cell exhaustion, whereas 4-1BB signaling induces mitochondrial biogenesis, in vivo T cell persistence, and reprogramming towards a central memory T cell phenotype [6]. However, there is limited information comparing these co-stimulatory domains in CAR T cells treating solid tumors. The design of CARs has followed several upgrades from a unique antigen-directed targeting to designed “living factories” delivering additional molecules to expand the antitumor purpose of CAR T cells [7]. For solid tumors, the potential of these factories must be fully explored to overcome the detrimental consequences induced by the TME, such as intense T cell exhaustion, suboptimal T cell trafficking, and non-homogeneous antigen expression.

The typical accelerated growth of solid tumors is not usually followed by similar synchronic levels of vascularization, resulting in tissue hypoxia guided by the hypoxia-inducible factor-1α (HIF-1α). The accumulation of HIF-1α causes the transcription of several genes involved in hypoxia response, including glycolytic enzymes, vascular endothelial growth factor, erythropoietin, and carbonic anhydrase IX (CAIX) [8]. CAIX is a metalloenzyme that regulates the intracellular and extracellular pH overexpressed in hypoxic tumors such as glioblastomas [9], triple-negative breast cancer [10], and colorectal cancer [11]. CAIX is also the main tumor-associated antigen overexpressed in ccRCC due to the frequent mutation of the tumor suppressor gene *von Hippel-Lindau* found in about 75% of ccRCC cases [12], which promotes hypoxia-independent expression of the HIF-1α-regulated genes, including CAIX [13,14]. Despite the great potential of CAIX for developing cancer-targeted therapies, the expression of this enzyme occurs in a few healthy tissues, such as intrahepatic biliary ducts [15], triggering hepatotoxicity in patients treated with anti-CAIX murine G250 CAR T cells in clinical trials [16,17]. The complete regression of clear cell renal cell carcinoma (ccRCC) obtained pre-clinically with newer anti-CAIX (humanized G36 clone) chimeric antigen receptor (CAR) T cells in doses equivalent to ≅10^8^ CAR T cells/kg in a CD4/CD8 mixture renewed the potential of this target to treat ccRCC and other hypoxic tumors [18,19].

Here, we compared the antitumoral preclinical efficacy of an intermediate dose of CD8α/4-1BB versus CD28-based anti-CAIX (G36 clone) CAR T cells providing an immune booster by releasing anti-PD-L1 antibodies against ccRCC. We have also checked for potential liver and renal toxicity induced by these CAR T cells, which have potential applications for treating ccRCC and other CAIX+/PD-L1+ tumors.

## 2. Results

### 2.1. Functional Characterization and Cytotoxic Activity In Vitro of Anti-CAIX CAR T Cells CD28ζ versus CD8α 4-1BBζ Releasing Anti-PD-L1

The following second-generation CARs containing CD8 alpha/4-1BB/CD3ζ were constructed by molecular cloning: anti-CAIX/ZsGreen, anti-B cell maturation antigen (BCMA)/ZsGreen, and anti-CAIX/anti-PD-L1 stabilized IgG4 [20]. Such constructs were also compared to the previously produced and tested anti-CAIX CAR/CD28/anti-PD-L1 stabilized IgG4 [21]. All constructs had their complete sequences confirmed by Sanger sequencing.

The lentiviruses were produced and concentrated as described in the Methods section. The viral titer obtained ranged from 10^7^–10^8^ TU/mL. The peripheral blood mononuclear fraction (PBMCs) was purified and maintained in the presence of IL-7 and IL-15. In Figure 1, we can note that CAR T cells showed proliferation in vitro, reaching 75–97% transduction levels four days after transduction with the lentiviruses (Figure 1A–C) and maintaining about 40% transduction after 14 days (Figure 1D). Half a million T cells/mL secrete about 350 ng/mL of anti-PD-L1 IgG4 after two days of incubation (Figure 1E), representing circa 0.35 pg/cell/day. We performed all cytotoxicity assays using skrc59 80% double positive for CAIX and PD-L1, with almost 20% of CAIX negative cells, from which 15% were only positive for PD-L1. We decide to challenge these cells without resorting them to CAIX or PD-L1 to see the performance of our anti-CAIX CAR T cells in a non-homogeneous setting of CAIX/PD-L1 expression representing more realistically the heterogeneous populations of cells usually present in the human ccRCC microenvironment. Regarding the in vitro antitumor effect on Skrc59 CAIX+/PD-L1+ human ccRCC, we found that all anti-CAIX CAR T cells had a higher cytotoxic activity when compared to the negative control (anti-BCMA CAR), independent of the CD28 or 4-1BB co-stimulatory domain or the secretion of anti-PD-L1 IgG4 at a 25:1 effector cell/tumor cell ratio (E:T) when treated for 24 h (Figure 1F). Higher E:T (50 or 100:1) showed even more potent results, reaching up to 80% cytotoxicity of the same ccRCC cells. These CAR T cells could not induce the cytotoxicity of CAIX negative cells, as previously tested [18,21].

### 2.2. Exhaustion Status of Anti-CAIX CAR T Cells CD28ζ versus CD8α 4-1BBζ Releasing Anti-PD-L1 IgG4 Antibodies in Co-Culture with Human ccRCC Cells In Vitro

To assess whether the secretion of anti-PD-L1 IgG4 secreted by the anti-CAIX CAR T cells would be able to reverse T cell exhaustion, we first performed an in vitro exhaustion assay. The CAR containing the CD28 co-stimulatory domain was unique to increase in 40% of the population of live T cells that does not express any of the following exhaustion markers: programmed cell death receptor (PD-1), T cell immunoglobulin, and mucin domains (TIM-3) or cytotoxic T lymphocyte-associated protein 4 (CTLA-4), associated with a 15% increase in interleukin-2 (IL-2) expression compared to the negative control anti-BCMA CAR (Figure 1G). The anti-CAIX/anti-PD-L1 IgG4 construct with 4-1BB significantly reduced the T cell population expressing all of the exhaustion markers analyzed; however, in terms of the absolute number, CAR T cells expressing all exhaustion markers were relatively rare in all groups. The percentage of IL-2 positive cells, on the other hand, increased circa 15% in all anti-CAIX CAR T groups compared to the negative control anti-BCMA CAR.

### 2.3. Comparative Evaluation of Anti-CAIX CAR T Cells CD28ζ versus CD8α 4-1BBζ Releasing Anti-PD-L1 IgG4 Antibodies in an Orthotopic NSG Mice Model of Human ccRCC

In Figure 2A, we can see an expressive reduction in the tumor size in groups treated with anti-CAIX CAR T capable of releasing anti-PD-L1 IgG4 antibodies. Furthermore, we observed the presence of metastases in the spleen and peritoneum of two out of four mice treated with non-transduced lymphocytes. In the group treated with anti-BCMA CAR T cells, we have found metastases in four out of five treated mice, with macroscopical foci visible in the liver, spleen, peritoneum, and diaphragm. Mice treated with anti-CAIX/4-1BB/ZsGreen CAR T cells also presented metastases in four out of five animals in the same locations, but the lesions were smaller and less vascularized (Figure 2A). In the group treated with anti-CAIX/4-1BB capable of releasing anti-PD-L1 IgG4, we found metastases in two out of five animals, all restricted to the peritoneum, diaphragm, and bladder. In the group treated with anti-CAIX/CD28 releasing anti-PD-L1 IgG4, we found only local tumor advance, with tiny lesions restricted to the diaphragm of only one out of five animals.

Regarding the tumor weight (Figure 2B), we observed a 30% reduction in all mice treated with anti-CAIX 4-1BB CAR T cells with or without the expression of anti-PD-L1 antibodies and a 60% reduction upon treatment with anti-CAIX CD28 CAR T cells capable of expressing anti-PD-L1 IgG4 antibodies. Fragments of the tumor tissue were used to (i) evaluate exhaustion markers expressed by tumor-infiltrating lymphocytes (TILs) (Figure 2C) and (ii) prepare formalin-fixed, paraffin-embedded tumor tissues for immunohistochemistry using antibodies for Ki67, PD-L1, and CD3 and stain with hematoxylin-eosin. Regardless of the co-stimulatory domain or the anti-PD-L1 IgG4 secretion, all mice treated with anti-CAIX CAR T cells showed a 50% reduction in the co-expression of the exhaustion markers PD-1, TIM-3, CTLA-4, and CD39 in TILs, as determined by flow cytometry. Figure 2D shows the percentage of positive cells and the median expression intensity of each exhaustion marker individually. In this case, we found a possible direct effect of the CAR anti-CAIX 4-1BB blocking T cell exhaustion since there was no significant difference in the expression of the exhaustion markers between this construction and the one with 4-1BB that releases anti-PD-L1 IgG antibodies.

We have also evaluated the CAR T cell expression and levels of T cell exhaustion markers (CD39, CTLA-4, TIM-3, and PD-1) on circulating CAR T cells collected from whole blood before euthanasia (Figure 3). All groups treated with Anti-CAIX CAR T cells, regardless of the type of co-stimulatory domain or the secretion of Anti-PD-L1 IgG4, showed a 50% reduction in the percentage of circulating CD8 positive T cells expressing all exhaustion markers when compared to the groups treated with the control CAR T cells (Figure 3A). Circulating CD4 T cells showed a significant reduction of approximately 40% of cells positive for all exhaustion markers only in groups capable of secreting Anti-PD-L1 IgG4 (Figure 3B). Few live CD8 (Figure 3A) and CD4 (Figure 3B) CAR T cells were found circulating in the peripheral blood one month after T cell transfer, and the population of CD4+ cells in the group of both Anti-CAIX CAR T cells containing 4-1BB was higher compared to the negative anti-BCMA control. About 65% of the CD4 or CD8 T cells found in the blood were positive for the CAR (Figure 3A–CD8 and Figure 3B–CD4).

The immunohistochemistry of the tumors from mice treated with anti-CAIX/anti-PD-L1 IgG CAR T cells containing 4-1BB and CD28 revealed increased CD3 infiltration (Figure 4A,B) and decreased expression of both PD-L1 (Figure 4A,C) and the proliferation marker Ki67 (Figure 4A,D,E), this being especially relevant for the CD28 construction compared to all other groups.

To assess the potential liver toxicity induced by the on-target off-tumor effect of anti-CAIX CAR T cells in the biliary duct, we analyzed the infiltration of CD3 cells in the liver by immunohistochemistry (Figure 5A,B); however, no statistical differences in the infiltration of anti-CAIX CAR T cells in the hepatic parenchyma among the groups were found after image quantification. (Figure 5B). We also measured alanine transaminase (ALT), aspartate transaminase (AST) (Figure 5C), and creatinine (Figure 5D) levels in the plasma of mice to assess liver and kidney function [23]. No significant difference in the expression of these markers was found among the groups, providing further evidence for the absence of hepatic and nephrotoxicity with doses equivalent to 10^7^ anti-CAR T cells/kg.

## 3. Discussion

CAIX was one of the first targets developed for CAR T cell therapy in ccRCC. Therefore, researchers had not enough prior clinical experience to determine the appropriate conditions to perform this type of treatment, and the first phase I clinical trial performed in 12 patients with ccRCC applied a first-generation murine anti-CAIX CAR T (G250mAb) CD4_TM_-γ CAR, with daily infusions of 2 × 10^8^ to 2 × 10^9^ of anti-CAIX CAR T in association with IL-2 and with a maximum of 10 sequential infusions, obtaining disappointing results in terms of toxicity and efficacy. Patients developed anti-CAR T antibodies and immune responses that led to degrees of hepatotoxicity from two to four due to the physiological expression of CAIX found in this tissue, with four out of eight patients having to discontinue the treatment. No objective response was detected [16,17,24]. With the current knowledge, failure of this protocol would be expected, as sequential daily doses of murine CAR T cells associated with IL-2 would induce a massive but not long-lasting immune response, as first-generation CAR T cells are known for their low sustained maintenance [25].

Due to the relevance of CAIX as a tumor-associated antigen expressed in most cases of ccRCC and associated with low expression in non-tumor tissues and the design biases of the aforementioned clinical study, we continue to work on the development of other generations of humanized anti-CAIX CARs containing extracellular, transmembrane, and intracellular domains of CD28, which were superior to the first generation concerning the objective response observed in an orthotopic NSG mouse model of human ccRCC but still did not fulfil the expectations and require improvements and optimizations [26]. In a subsequent article, the same humanized anti-CAIX/CD28 lentivector was adapted to secrete anti-PD-L1 IgG1 or IgG4 antibodies into the tumor milieu, which led to a remarkable reduction of T cell exhaustion and improved antitumor persistence in an orthotopic model of ccRCC in NSG mice. Only IL-21 treated CD8 T cells were adoptively transferred in that study. Despite the excellent objective response observed, it was extremely challenging to maintain long-lasting CAR T cells in the blood circulation [21]. Knowing the importance of CD4 T cells in the maintenance of CD8 T cells and the ability of IL-7 and IL-15 cytokines to induce a central memory T cell phenotype when used in vitro for CAR T cell expansion, we performed this study comparing anti-CAIX CAR constructs capable of promoting the release of anti-PD-L1 stabilized IgG4 containing 4-1BB or CD28 as co-stimulatory domains.

The in vitro data revealed that all anti-CAIX CAR T cells could induce the cytotoxicity of CAIX+/PD-L1+ skrc59 cells, with more potent results obtained when higher E:T ratios were applied. In 2015, we used CAIX/PD-L1 skrc59 cells in our co-culture assays after sorting for CAIX+PD-L1+ skrc59 cells. Currently, this lineage presents 80% of cells positive for both CAIX and PD-L1, representing a more realistic non-homogeneous profile of CAIX and PD-L1 expression frequently found in the human ccRCC microenvironment. This profile explains the lack of complete antitumoral activity in vitro, which was observed even when higher E:T ratios were used.

The results obtained in the in vivo orthotopic ccRCC model showed that anti-CAIX CAR T cells that release anti-PD-L1 promoted a significant reduction in tumor volume and weight, with the construction with CD28 showing more potent results compared to 4-1BB, preventing the induction of tumor metastases. Additionally, the construction with CD28 was extremely efficient in avoiding tumor dissemination, as only one animal had a tumor with localized invasion of the diaphragm, and diminished PD-L1 and Ki67 expression by the tumor. Considering T cell exhaustion, the constructions with anti-CAIX CD28 CAR and anti-PD-L1 secretion and with 4-1BB CAR with or without anti-PD-L1 secretion showed reduced co-expression of PD-1, TIM-3, CTLA-4, and CD39 in viable tumor-infiltrating T cells. The downregulation of immune checkpoint gene expression for anti-CAIX G36 scFv 4-1BB was also observed and detailed in a recently published article [18], suggesting that 4-1BB itself may induce some immunological checkpoint blockade. This effect did not occur for CD28, since previously published results showed that anti-CAIX CD28 CAR T cells without anti-PD-L1 release did not significantly reduce T cell exhaustion [21].

All of the anti-CAIX CAR T cells tested in this paper induced a significant decrease of CD39 expression. CD39 is an ectoenzyme that binds ATP and, in association with CD73, converts it to extracellular adenosine [27]. Adenosine has several intense immunosuppressive effects when it acts on it’s a2AR and A2BR receptors, mainly by increasing cyclic AMP. The high levels of cAMP signaling promote the suppression of CD8 T cell effector functions, such as the production of pro-inflammatory cytokines, proliferation, and cytotoxic activity [28]. CD39 also enhances the immunosuppressive effects of regulatory CD4+CD25+ forkhead box P+ (FOXP3) T cells [29]. Wang et al. have found superior tumor-infiltrating regulatory T cells in ccRCC treated with anti-CAIX CD28 CAR T cells without an immune checkpoint blockade compared with anti-CAIX 4-1BB CAR T cells. Here, we note that ICB via anti-PD-L1 released by anti-CAIX CD28 CAR T cells achieves similar levels of CD39 expression compared with 4-1BB-based anti-CAIX CAR T cells with or without ICB via PD-L1, suggesting a possible synergic effect of CD-28 based CARs with a PD-L1 blockade regarding immunosuppression reversal [18]. Particularly for solid tumors, the identification of a co-stimulatory CAR signal capable of inducing T cell escape from robust T_reg_-mediated inhibition associated with immune checkpoint blockades would be of paramount importance.

For the anti-CAIX CD28 CAR T cells releasing anti-PD-L1 mAbs, improving the CAR T culture conditions using IL-7 and IL-15 and full PBMCs allowed for similar results to those obtained previously with a dosage ten times higher than that applied here and the use of CD8 positive CAR T cells only [21], with low levels of exhaustion markers in circulating and infiltrating CAR T cells, the prevention of metastases, the permanence of circulating CAR T cells, and the absence of liver or renal toxicity. Moreover, the role of CD4+ T cell depletion in cancer and other diseases is still poorly understood [30], and our data evidenced that a decrease in the exhaustion markers of circulating CD4 T lymphocytes seems relevant to improving the antitumoral function of CAR T cells. The superior antitumoral and antimetastatic effects observed for anti-CAIX CD28 CAR T cells secreting anti-PD-L1 mAbs compared to all other groups are possibly explained by the fact that PD-1 can inhibit the phosphorylation of CD28 [31], and the blockade of PD-L1 with anti-PD-L1 mAbs secreted by CAR T cells with the consequent inactivation of PD-1 signaling might restore CD28 function in T cells. Since the CD28-based anti-CAIX CAR T cells provide T cells with more CD28 expression while releasing anti-PD-L1 IgG, this can potentiate even more CAR T cell co-stimulation, explaining the superiority of CD28 versus 4-1BB co-stimulation only when associated with anti-PD-L1 secretion. The lower PD-L1 and Ki67 expression observed in tumors from mice treated with anti-CAIX CD28 anti-PD-L1 IgG compared with the anti-CAIX 4-1BB anti-PD-L1 IgG CAR T cells may be explained by the superior cytotoxicity of PD-L1 positive ccRCC cells induced by the improvement of CD28-based anti-CAIX CAR T cells function by anti-PD-L1 IgG. The comparison between anti-CAIX CD28 with or without anti-PD-L1 mAbs secretion using high doses to treat ccRCC was previously published [21].

A CAR construction based on the same anti-CAIX G36 scFv 4-1BB but with no anti-PD-L1 secretion was recently used to produce CAR T cells in a CD4/CD8 ratio of 2:1, leading to complete ccRCC remission in an orthotopic model in NSG-SGM3 mice at a dose equivalent to ≅10^8^ CAR-T cells/kg dose, with mice remaining tumor-free 72 days after infusion. This powerful treatment was able to downregulate immune checkpoint genes and reduce the differentiation of regulatory CD8 T cells. Hepatotoxicity or other toxicities were not investigated in this study [18]. The outstanding preclinical results of anti-CAIX G36 scFv 4-1BB CAR T cells alone were superior to those of their CD28-based counterpart. This guided us to test similar CAR T cells with an additional capacity of secreting anti-PD-L1 mAbs, using two interspaced injections of CAR T cells in a dose four times lower to avoid a potential hepatoxicity while keeping or improving its functioning by ICB. Despite the absence of hepatotoxicity, the lower doses of 4-1BB-based CAR T cells were not as potent as higher doses tested [18], not even when CAR T cells released anti-PD-L1 mAbs. Conversely, CD28-based anti-CAIX CAR T cells releasing anti-PD-L1 induced more potent antitumoral effects than the 4-1BB counterpart. When the PD-L1 immune checkpoint blockade is associated with CAR T cells, CD28 seems to be a preferential co-stimulatory domain to include in the CAR structure, since blocking the PD-1/PD-L1 axis boosts the signaling of CD28 highly expressed on such CAR T cells. PD-1/PD-L1 signaling blockage is not able to directly modulate 4-1BB signaling, which could explain the similar results obtained between the anti-CAIX 4-1BB CAR T cells capable or not capable of secreting anti-PD-L1, which differ only in terms of the superior antimetastatic properties for the CAR T releasing anti-PD-L1, reinforcing that 4-1BB itself may exert some immunological checkpoint blockade. Moreover, the CD4/CD8 CAR T cells ratio of 2:1 cultured initially with IL-21 and then expanded with IL-7 and IL-15 seems crucial to enhance the performance of anti-CAIX CAR T cells [18]. Moreover, it is critical to consider that the inherent immunosuppression of the NSG mouse model used to develop human tumors probably underestimated the CAR T antitumoral effect, leading us to believe that even lower CAR T cell doses have the potential to achieve complete remission in non-immunocompromised humans.

Several recently published CAR T clinical trials applied doses ranging from 10^6^ to 10^8^ CAR T cells/kg in the patients, most of them applying a scheme of one or two injections with dose-dependent toxicity [32,33,34,35]. The dual anti-CAIX/anti-PD-L1 cellular therapy applied in an intermediate dose of 10^7^ CAR T cells/kg presented no detectable hepatic or renal toxicity in mice. It remains to be elucidated if the most effective higher doses of anti-CAIX CAR T cells, such as the equivalent to 10^8^ cells/kg previously tested, would cause relevant hepatotoxicity [18].

Given the promising pre-clinical results recently published [18] and the ones presented herein proving a lack of toxicity with intermediate doses of anti-CAIX CAR T cells, clinical studies based on these adoptive cell therapies with doses superior to 10^7^ CAR T cells/kg are recommended, using anti-CAIX CD28 CAR T cells releasing anti-PD-L1 antibodies or anti-CAIX 4-1BB CAR T cells, offering exciting new prospects for the treatment of refractory ccRCC and hypoxic tumors.

## 4. Materials and Methods

### 4.1. Cell Lines and Culture

The human clear cell renal carcinoma cell line skrc59 CAIX+/PD-L1+ was kindly provided by Dr. Marasco (Dana-Farber Cancer Institute, Boston, MA, USA). These cells were cultured in RPMI 1640 medium (Life Technologies, Carlsbad, CA, USA) supplemented with 10% (*v*/*v*) FBS (Sigma-Aldrich, San Louis, MO, USA), 100 IU/mL penicillin, and 100 µg/mL streptomycin (Sigma-Aldrich, San Louis, MO, USA). T cells were maintained in a complete RPMI medium containing 20 mM HEPES and 10 ng/mL interleukin 7 (IL-7) and interleukin 15 (IL-15) (Peprotech, Rocky Hill, NJ, USA), which were added to the medium every other day. 293T (CRL-11268, ATCC) and Lenti-X 293T (Clontech, Mountain View, CA, EUA) cells were maintained in DMEM with 10% FCS, 100 IU/mL penicillin, and 100 µg/mL streptomycin (Sigma-Aldrich, San Louis, MO, USA). Skrc59 cells were modified to highly express human CAIX, and CAIX+/PD-L1+ cells were selected by sorting (FACSAria, BD Biosciences, Franklin Lakes, NJ, USA). All lines were maintained at 37 °C, 5% CO_2_.

### 4.2. Cloning of a CD8 Alpha Spacer and the 4-1BB Co-Stimulatory Domain into a Bicistronic Lentiviral Vector Encoding an Anti-CAIX CAR

We synthesized the coding DNA sequence for the extracellular portion of CD8α associated with the transmembrane and intracellular portion of the co-stimulatory molecule 4-1BB and CD3ζ (Genewiz, South Plainfield, NJ, USA). We then cloned this sequence to replace the spacer region previously occupied by the C9 tag, CD28/CD3ζ, between the 5′ NotI and 3′ PacI restriction sites of the pHAGE vector. The stabilized monoclonal anti-PD-L1 IgG4 coding DNA sequence (11A clone) [20] had its codons optimized and was synthesized (Genewiz, South Plainfield, NJ, USA) containing the 5′ NdeI and 3′ ClaI restriction sites and subcloned in place of ZsGreen in the second expression cassette of this anti-CAIX containing lentivirus (clone G36) scFv/CD8α/4-1BB/CD3ζ. As a negative control, ZsGreen was cloned in the second cassette of this vector, after IRES. For the negative control of CAR, the anti-CAIX scFv part was replaced by an anti-B lymphocyte maturation antigen (BCMA) scFv. These clonings resulted in three constructions for the production of lentiviruses: anti-CAIX CAR 4-1BB capable of expressing ZsGreen (anti-CAIX/4-1BB/ZsGreen), anti-CAIX CAR 4-1BB capable of expressing anti-PD-L1 IgG4 (anti -CAIX/4-1BB/anti-PD-L1 IgG), and anti-BCMA 4-1BB CAR capable of expressing ZsGreen (anti-BCMA/4-1BB/ZsGreen), which were compared with anti-CAIX CAR CD28 capable of expressing anti-PD-L1 IgG4 (anti-CAIX/CD28/anti-PD-L1 IgG).

### 4.3. Lentiviral Production

Lentiviruses were produced by the transient transfection of five plasmids into 293T cells using polyethyleneimine (PEI) (Sigma-Aldrich, San Louis, MO, USA) [36]. Briefly, each 15 cm plate containing 80% confluent 293T LentiX was transfected with 30 µg of the five plasmids in total—this being 5 µg of each of the following structural plasmids: pHDH-Hgpm2 (HIV gag-pol), pMD-tat, pRC/CMV-rev, and Env VSV-G—and 10 µg of the plasmid coding for the different types of CAR. The virus-containing supernatants were concentrated using a final concentration of 8.5% polyethylene glycol 6000 and 0.3M sodium chloride diluted in PBS, as previously described [37], and kept frozen at −80 °C.

### 4.4. Selection, Activation, and Transduction of PBMCs

Whole blood from healthy volunteer donors was collected after the donors signed an informed consent form. Peripheral blood mononuclear fraction (PBMCs) was separated using Ficoll-Paque PLUS (GE Healthcare, Chicago, IL, USA). The Human T-Activator CD3/CD28 Dynabeads (Life Technologies, Carlsbad, CA, USA) were used in a 1:1 ratio for the activation and expansion of lymphocytes and maintained in IL-7 and IL-15 10 ng/mL (Peprotech, Rocky Hill, NJ, USA), which promote the in vivo expansion of CAR T cells with a predominant phenotype of memory stem cell (CD8+CD45RA+CCR7+) that produces more significant antitumor activity, with greater persistence and preservation of their migratory capacity to secondary lymphoid organs [38]. Subsequently, the cells were transduced with lentivirus in the multiplicity of infection of 6 and 10 µg/mL of diethylaminoethyl. All assays were performed in triplicate, and the T cells were isolated from three healthy donors.

### 4.5. T Cell Transduction and Payload Secretion Analysis

Cell transduction was confirmed by analyzing ZsGreen or cells stained with 10 µg/mL of human CAIX-Fc or human BCMA-Fc (Ab Biosciences, Concord, MA, USA) and then incubated with 2 µg/mL of mouse IgG anti-human IgG or rabbit anti-mouse IgG conjugated to APC (ThermoFisher, Waltham, MA, USA). CountBrightTM Absolute Counting Beads (ThermoFisher, Waltham, MA, USA) were used for proliferation analysis. The samples were analyzed by flow cytometry, and the data were analyzed using the FlowJo program. The total level of IgG secreted into the medium by cells transduced with the CAR-coding lentiviruses was determined using the “IgG Human Uncoated ELISA kit” (ThermoFisher, Waltham, MA, USA) using MaxiSorp 96-well plates (Nunc, Rochester, NY, USA).

### 4.6. Cytotoxic Effect of Anti-CAIX CAR T Cells Producing Anti-PD-L1 Antibodies on the Renal Carcinoma Cells

First, 2.5 × 10^3^ CAIX+/PD-L1+ skrc59 cells were plated in 96-well plates for 24 h at 37 °C, 5% CO_2_. Four days after T cell transduction, CAR T cells were added at 25:1, 50:1, and 100:1 effector cells/tumor cells (E:T) ratios and incubated for 24 h at 37 °C, 5% CO_2_. Lactate dehydrogenase activity was measured in the supernatant (Bioclin, Sao Paulo, SP, Brazil).

### 4.7. CAR T Cell Exhaustion Status in Co-Culture with ccRCC Cells

CAR T cells were activated and cultivated for two days with Dynabeads anti-CD3/CD28 and 10 ng/mL of IL-7 and IL-15, followed by co-culture with CAIX+/PD-L1+ skrc59 cells for another four days to induce exhaustion. The expression of the exhaustion markers PD-1, TIM-3, and CTLA-4, as well as interleukin 2 (IL-2), was determined by flow cytometry within gated CD45+ live single cells. We first defined the single-cell population in the bisector region of an FSC-A versus FSC-H dot plot to establish the gates. Then, we defined the negative population of live/dead stained single cells (live cells). Inside the live single cells gate, we define a gate for CD45 positive cells (or CD4 or CD8 in the experiment of peripheral CAR T cells). T cell exhaustion markers or IL-2 expression were analyzed inside the CD45 (or CD4 or CD8, when applicable) positive live single cells. Co-expression of the exhaustion markers was also determined in the CD45+ live single cell population by Boolean analysis using FlowJo Software (FlowJo LLC, Ashland, OR, USA).

### 4.8. Comparative Evaluation of the Antitumor Efficacy and Exhaustion of Anti-CAIX-CAR T Cells Releasing Anti-PD-L1 Antibodies, Constructions 4-1BB versus CD28, in an Orthotopic Model of Clear Cell Renal Carcinoma

Twenty-five male NOD.Cg-Prkdcscid/Il2rgtm1Wjl/SzJ (or NSG) mice, 6–8 weeks old and raised in the animal facility of the International Research Center (CIPE) of the A.C Camargo Cancer Center, were used in this project, which was approved by the animal ethics committee (Process 088/21). For the implantation of the renal tumor, the animals were anesthetized with an intraperitoneal injection of Ketamine 100 mg/kg (Ketamin, Cristália, Itapira, SP, Brazil) and Xylazine 10 mg/kg (Anasedan, Ceva, Paulínia, SP, Brazil) and submitted to a left longitudinal lumbotomy of approximately 1 cm, and the kidney was accessed and isolated. Human kidney cells skrc59 CAIX+/PD-L1+ (5 × 10^4^) were prepared in 20 µL of culture medium diluted 1:1 in Geltrex™ (Life Technologies, Carlsbad, CA, USA) and injected into the renal capsule with a syringe after surgical displacement of the renal capsule with a capillary tube. The animals were injected with analgesic 0.1 mg/kg buprenorphine intraperitoneally and monitored until complete recovery from anesthesia in a heated blanket. One week after tumor implantation, 3 × 10^7^ CAR T+ cells/kg were injected into the tail vein of mice, and this procedure was repeated after fourteen days. Several recently published CAR T cells-based clinical trials have used doses ranging from 10^6^ to 10^8^ CAR T cells/kg, most applying one or two injections [32,33,34,35]. Considering the average weight of 26.5 g for male NSG mice at 6-8 weeks old [39], the dose of 7.5 × 10^5^ CAR T cells injected in mice represents a usually well-tolerated intermediate dose of approximately 3 × 10^7^ cells/kg [32,33,34,35], The assay was designed with N = 5 animals per group, but one animal of the control group died from the surgery. After 30 days from the start of treatment, we evaluated the capacity to reduce tumor volume and weight by the different anti-CAIX CAR T cells and the capacity to inhibit the T cell exhaustion of TIL and circulating CD4 and CD8 lymphocytes by flow cytometry. For euthanasia, the mice were subjected to deepening anesthesia with a cocktail of 85 mg/kg of ketamine associated with 8 mg/kg of xylazine, had their blood removed by cardiac puncture, and were transferred to a tube with citrate.

#### 4.8.1. Hepatic and Renal Toxicity

Alanine (ALT) and aspartate (AST) transaminases activity was determined by spectrophotometry (Cobas 8000, Roche/Hitachi, Basel, Switzerland) in the plasma to assess possible liver toxicity from the treatment of mice with each one of the anti-CAIX CAR T cells compared with the anti-BCMA CAR T cells or the untransduced T cells. We have also measured creatinine to assess kidney function by spectrophotometry (Cobas 8000, Roche/Hitachi, Basel, Switzerland) in the same samples. The spectrophotometric measurements considered the standards available in the respective commercial kits (Cobas 8000, Roche/Hitachi, Basel, Switzerland).

#### 4.8.2. Tumor Infiltrated CAR T Cells Assessment

The buffy coat was removed from the blood, and the circulating T cells were marked with live/dead and conjugated antibodies for CD45 (1:20, BD Biosciences, Cat. No. 566041, CD4 (1:20, BD Biosciences, Cat. No. 557852, CD8 (1:20, BD Biosciences, Cat. No. 565310), the exhaustion markers CD39 (1:20, BD Biosciences, Cat. No. 562794), CTLA4 (1:5, BD Biosciences Cat. No. 555853), TIM-3 (1:20) BD Biosciences Cat no. 565559), and PD-1 (1:20, BD Biosciences, 561272), and the CAR (stained with 10 µg/mL of human CAIX-Fc or mouse BCMA-Fc and then incubated with 2 µg/mL of mouse IgG anti-human IgG or rabbit anti-mouse IgG conjugated to APC) by flow cytometry. After weight, the tumors were divided into two parts. One of them was fragmented into small pieces, followed by digestion in 900 µL of RPMI 1640 medium with 50 µL of collagenase type I and type IV 4000 U/mL and 1 µL of DNAse 1.0 mg/mL, and filtered in a cell strainer to assess the presence of CAR expression in TIL, and the expression of T cell exhaustion markers (CD39, CTLA4, TIM3, PD-1) in CD45 gate and live cells was performed by flow cytometry. The other parts of the tumor and the liver were fixed in 10% formaldehyde in PBS for immunohistochemical analysis.

#### 4.8.3. Immunohistochemistry (IHC)

Fixed tumors and livers were embedded in paraffin, sectioned four micrometers thick, dewaxed, and rehydrated in a decreasing ethanol series. Endogenous peroxidase activity was quenched using 3% hydrogen peroxide. The antigen retrieval was performed in citrate buffer (pH = 6.0) for 45 s at 123 °C, 15 PSI. Tumor tissues were incubated for 45 min with anti-human PD-L1 (Clone MIH1 ThermoFisher 14-5983-82; 1:25), anti-human Ki67 (BD 550609, 1:25), and anti-human CD3 (BD 566685; 1:250), followed by incubation with peroxidase-conjugated secondary antibodies (Dako). The slides were stained with 3,3′-diaminobenzidine (DAB) with hematoxylin counterstain. The images were obtained in an Olympus BX51 microscopy using a DP71 digital camera (Olympus) and analyzed in the DP Controller Software (Olympus). The quantification of IHC images was performed using the “IHC Profiler Plugin” of the ImageJ program (23). Liver slides from the animals were labeled for CD3, in addition to hematoxylin/eosin (HE) labeling to assess possible hepatotoxicity due to the liver infiltration of CAR T cells.

### 4.9. Statistical Analysis

The statistical significance of the data was evaluated using GraphPad Prism version 7.00 for Windows (La Jolla, CA, USA). The variables were analyzed for non-parametricity using the Kolmogorov–Smirnov test. The Mann–Whitney test was used to determine the relationship between non-parametric variables and the t-test was used for parametric variables, where two groups were compared. When three or more groups were compared, the ANOVA test was applied for parametric variables associated with a Tukey posthoc test, and for non-parametric variables, the Kruskal–Wallis test was applied with a Dunn’s posthoc test. A significance level of 0.05 was considered.

## Figures and Tables

**Figure 1 ijms-23-05448-f001:**
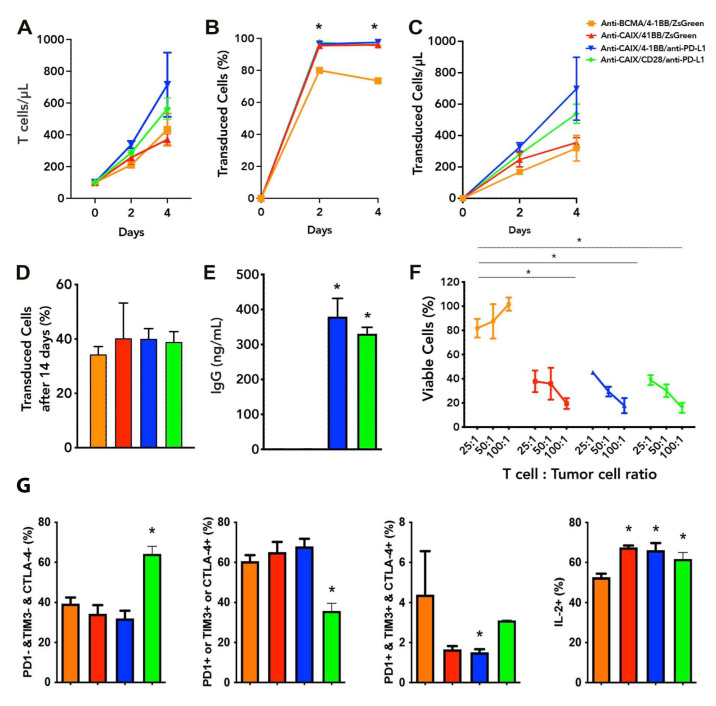
Functional characterization of chimeric antigen receptor (CAR) T cells and CAR T cell exhaustion in vitro. (**A**) T cells proliferation analysis two and four days after transduction with Anti carbonic anhydrase IX (CAIX) CD8alpha/4-1BB CAR-expressing anti-programmed cell death ligand-1 (PD-L1) IgG4 (anti-CAIX/4-1BB/anti-PD-L1 IgG4), or ZsGreen (anti-CAIX/4-1BB/ZsGreen), anti CAIX CD28 CAR-expressing anti-PD-L1 IgG4 (anti-CAIX/CD28/anti-PD-L1 IgG4) compared to anti-B cell maturation antigen (BCMA) 4-1BB CAR-expressing ZsGreen (anti-BCMA/4-1BB/ZsGreen negative control). (**B**) Percentage of CAR+ T cells and (**C**) Proliferation of CAR+ T cells two and four days after transduction. (**D**) Percentage of CAR T cells fourteen days after initial transduction, representing the long-term stable expression of CAR by integration of lentiviruses into the T cell genome. (**E**) IgG secretion by anti-CAIX/anti-PD-L1 IgG4 CAR T cells determined by ELISA (Sigma-Aldrich). (**F**) Cytotoxicity of Skrc59 CAIX+/PD-L1+ renal carcinoma cells promoted by CAR T cells in different proportions of effector cells per tumor cells (E:T). The results presented represent the mean ± SD of CAR T cells from three donors in triplicate. (**G**) CAR T cell exhaustion in vitro. CAR T cells were activated with anti-CD3/CD28 beads and cultured for five days with 10 ng/mL of IL-7 and IL-15, followed by co-culture with skrc59 CAIX+/PD-L1+ cells for another four days for induction of exhaustion. The expression of the exhaustion markers PD-1, TIM-3, CTLA-4, as well as interleukin 2 (IL-2) were determined by flow cytometry within live CD45 positive-gated cells. Co-expression of exhaustion markers was determined by Boolean analysis in the FlowJo Software (FlowJo LLC, Ashland, OR, USA). * *p* < 0.05 compared to anti-BCMA/ZsGreen CAR T cells.

**Figure 2 ijms-23-05448-f002:**
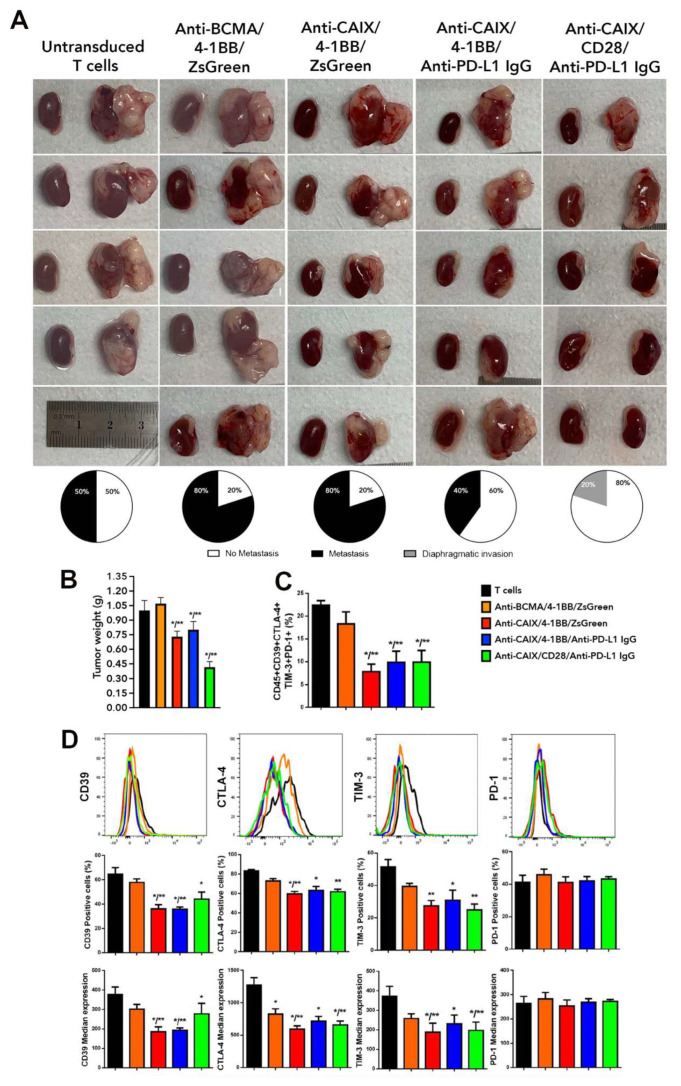
Comparative antitumor effect of anti-carbonic anhydrase IX (CAIX) chimeric antigen receptor (CAR) T cells CD28ζ versus CD8α 4-1BBζ releasing anti-programmed cell death ligand-1 (PD-L1) IgG4 antibodies in an orthotopic model of human clear cell renal carcinoma (ccRCC). (**A**) Images of the kidneys with and without tumor of each of the mice treated with non-transduced T cells (T cells), anti-B cell maturation antigen (BCMA)/4-1BB/ZsGreen CAR T cells, anti-CAIX/4-1BB/ZsGreen CAR T cells, anti -CAIX/4-1BB/anti-PD-L1 IgG4 CAR T cells, and anti-CAIX/CD28/anti-PD-L1 IgG4 CAR T cells (N = 5/group) derived from three different donors (one or two mice were injected with CAR T cells from each donor inside a group) and the percentage of mice that presented metastasis (black) or localized invasion (gray) within each group, respectively. One mouse died after surgery, and, for this reason, the group treated with untransduced T cells had N = 4. (**B**) Tumor weight was determined by subtracting the weight of the left cancerous kidney presenting a tumor by the weight of the respective healthy right kidney. (**C**) Boolean analysis of the co-expression of all tested T cell exhaustion markers (CD39, CTLA-4, TIM-3, and PD-1) in live tumor-infiltrating lymphocytes. (**D**) Flow cytometry histograms with graphs of the percentage of cells labeled alone for each exhaustion marker and individual median expression in viable TILs. The results were analyzed using the Kruskal–Wallis test applied with a Dunn’s posthoc test. The results represent the average ± SD. * *p* < 0.05 compared to untransduced T cells, ** *p* < 0.05 compared to anti-BCMA/ZsGreen CAR T cells.

**Figure 3 ijms-23-05448-f003:**
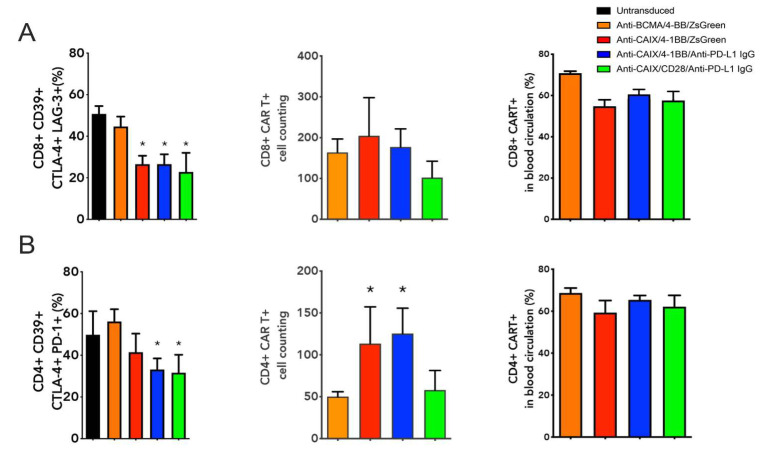
Immune profile of live peripheral CD8 and CD4 chimeric antigen receptor (CAR) T cells one month after the first T cell transfer. (**A**) Boolean analysis of all T cell exhaustion markers (CD39, CTLA-4, TIM-3, and PD-1) expressed by CD8 T cells; absolute cell count of CD8+ CAR T cells within gated live cells; and percentage of live peripheral CD8+ CAR T cells. (**B**) Boolean analysis of all T cell exhaustion markers (CD39, CTLA-4, TIM-3, and PD-1) expressed by CD4+ T cells; absolute cell count of CD4+ CAR T cells within gated live cells; and percentage of live peripheral CD8+ CAR T cells. Data determined by flow cytometry. * *p* < 0.05 compared to anti- B cell maturation antigen (BCMA) CAR.

**Figure 4 ijms-23-05448-f004:**
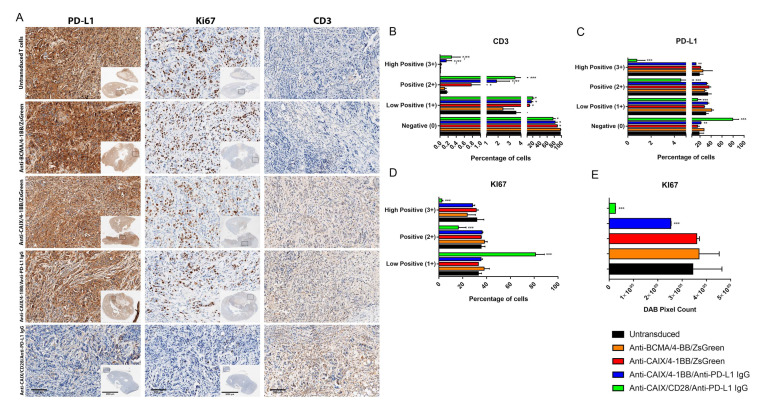
Immunohistochemical analysis of human clear cell renal cell carcinoma (ccRCC) from an orthotopic NSG mice model to evaluate in vivo activity of Anti-carbonic anhydrase IX (CAIX) CAR T cells. (**A**) Immunohistochemistry analysis of tumor sections by detecting PD-L1, KI67, and CD3 expression. The scale bars represent the magnification of the images of each column [2000 μm (4×) or 300 μm (20×)]. (**B**–**E**) Immunohistochemistry (IHC) Quantification. The images quantification was performed using the IHC Profiler Plugin of ImageJ Software [22]. The percentages of negative (0), low positive (1+), positive (2+), or high positive (3+) cells were shown. For CD3 (**B**) and programmed cell death ligand-1 (PD-L1) (**C**) quantification, cytoplasmatic quantification was applied where all of the image diaminobenzidine (DAB) pixels were counted. For the quantification of nuclear protein KI67 (**D**), the DAB staining pattern is confined to the nuclei, the threshold feature is used to select the positive-stained areas for quantification, and non-staining nuclei are not recorded [22]. DAB total pixels were also shown (**E**) to evaluate total Ki67 staining in all fields, including negative areas. * *p* < 0.05 compared with untransduced and anti-B cell maturation antigen (BCMA)/4-1BB/ZsGreen, ** *p* < 0.05 compared with anti-CAIX/4-1BB/ZsGreen, *** *p* < 0.05 compared with anti-CAIX/anti-PD-L1 IgG4 compared with all other groups.

**Figure 5 ijms-23-05448-f005:**
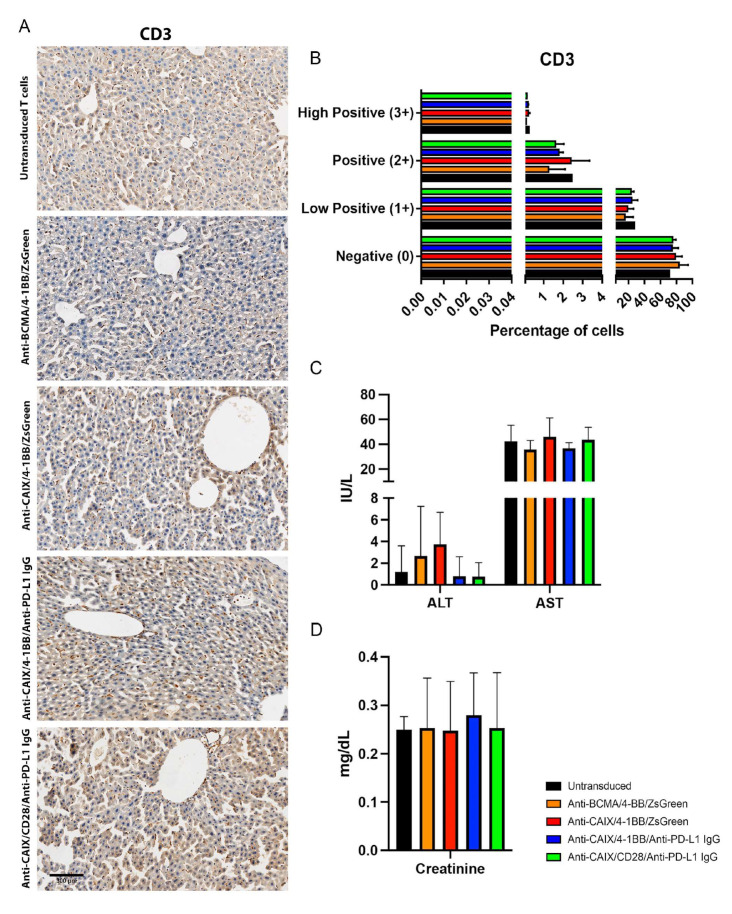
Evaluation of liver infiltrating CD3+ cells and plasmatic quantification of enzymes indicative of hepatic and renal injury in mice after chimeric antigen receptor (CAR) T cells therapy. (**A**) Immunohistochemistry (IHC) for CD3+ T cells detection in the liver. The scale bars represent the magnification of the images of each column [300 μm (20×)]. (**B**) CD3 Quantification. The quantification of the IHC images was performed using the IHC Profiler Plugin of ImageJ Software [22]. The percentages of negative (0), low positive (1+), positive (2+), or high positive (3+) cells were shown. (**C**) Alanine transaminase (ALT), aspartate transaminase (AST), and (**D**) creatinine plasmatic dosage in mice. The dosages were performed in the mice plasma by the spectrophotometric method (Cobas 8000, Roche/Hitachi) one month after the beginning of the therapy with two injections of 3 × 10^7^ CAR T cells/kg with fifteen days of interval between them.

## Data Availability

Not applicable.

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
