# Peer review of "Immune Checkpoint Blockade via PD-L1 Potentiates More CD28-Based than 4-1BB-Based Anti-Carbonic Anhydrase IX Chimeric Antigen Receptor T Cells"

_ijms, 2022, doi:10.3390/ijms23105448_

Round 1
Reviewer 1 Report
In this manuscript, the authors investigated the therapeutic effect of anti-carbonic anhydrase IX CAR T cells (Anti-CAIX/4-1BB/Anti-PD-L1 IgG) to clear cell renal carcinoma (ccRCC). The in vitro data revealed that the anti-CAIX CAR T cell can induce the cytotoxicity of CAIX+/ PD-L1+ skrc59 cells. Additionally, the releasing of anti-PD-L1 IgG4 significantly reduce the expression level of exhaustion markers. In the in vivo study, the authors demonstrated that anti-CAIX/4-1BB/Anti-PD-L1 IgG CAR T cells significantly reduce the growth and metastasis of skrc59 cells in NSG mice. The immune profile showed that the exhaustion makers were significantly reduced in anti-CAIX CAR T cells. Overall, the data presented in the manuscript revealed that the anti-CAIX/4-1BB/Anti-PD-L1 IgG CAR T cells have the potential for the treatment of ccRCC. Few comments are listed below.
- Figure 4A: The CAR T cells from Anti-CAIX/4-BB/Anti-PD-L1 IgG group and Anti-CAIX/CD28/Anti-PD-L1 IgG group can all release anti-PD-L1 IgG4 antibodies. However, according to the result obtained in immunohistochemistry, the PD-L1 level is significantly higher in Anti-CAIX/4-BB/Anti-PD-L1 IgG group than Anti-CAIX/CD28/Anti-PD-L1 IgG group. The authors may briefly discuss the possible reason in the manuscript.
- Figure 2: The author may need to describe how they do the statistical analysis in this figure.
- Figure 2D: It is suggested that the authors use a figure to show how they do the gating for flow cytometric analysis.
- Figure 3A and B, left panels: The authors may need to show what is the group of black bars shown in these figures.
- Line 151: A typo was found: “41BB”.
Reviewer 2 Report
The authors presented an interesting study following their previous research in the topic. Their results suggest the potential of using adoptive cell therapies for the treatment of refractory clear cell renal carcinoma though this might be dependent on an individual basis and a combination of parameters such as the use of different ILs. Have the authors investigated the use of IL-6 and/or IL-8 in their model? Also, more information is required in the Methodology for the controls used in the hepatic and renal toxicity.
As their research is based on the use of established cell lines, it would be interested to investigate the potential of cytotoxicity of CART cells in primary short-term cultures as more representative models of the primary disease.
